# A Neutralization Assay Based on Pseudo-Typed Lentivirus with SARS CoV-2 Spike Protein in ACE2-Expressing CRFK Cells

**DOI:** 10.3390/pathogens10020153

**Published:** 2021-02-02

**Authors:** Yalçın Pısıl, Hisatoshi Shida, Tomoyuki Miura

**Affiliations:** 1Laboratory of Primate Model, Research Center for Infectious Diseases, Institute for Frontier Life and Medical Science, Kyoto University, 53 Shogoin Kawahara-cho, Sakyo-ku, Kyoto 606-8507, Japan; yalcin.pisil@gmail.com; 2Graduate School of Human and Environmental Studies, Department of Interdisciplinary Environment, Dynamics of Natural Environment, Dynamics of Biological Environment, Kyoto University, Kyoto 606-8501, Japan; 3Division of Molecular Virology, Institute of Immunological Science, Hokkaido University, Sapporo 060-0808, Japan; hmyy2010@yahoo.co.jp

**Keywords:** SARS-CoV-2, Covid-19, HIV, CRFK Cell, ACE2, spike, monoclonal antibody, neutralization assay, pseudo-typed lentivirus

## Abstract

Severe acute respiratory syndrome coronavirus 2 (SARS-CoV-2) is a highly pathogenic zoonotic virus that spreads rapidly. In this work, we improve the hitherto existing neutralization assay system to assess SARS-CoV-2 inhibitors using a pseudo-typed lentivirus coated with the SARS-CoV-2 spike protein (LpVspike +) and angiotensin-converting enzyme 2 (ACE2)-transfected cat Crandell–Rees feline kidney (CRFK) cells as the host cell line. Our method was 10-fold more sensitive compared to the typical human embryonic kidney 293T (HEK293T) cell system, and it was successfully applied to quantify the titers of convalescent antisera and monoclonal anti-spike antibodies required for pseudo virus neutralization. The 50% inhibition dilution (ID50) of two human convalescent sera, SARS-CoV-2 immunoglobulin G (IgG) and SARS-CoV-2 immunoglobulin M (IgM), which were 1:350 (±1:20) and 1:1250 (±1:350), respectively. The 50% inhibitory concentration (IC50) of the IgG, IgM and immunoglobulin A (IgA) anti-SARS-CoV-2 monoclonal antibodies (mAbs) against LpVspike(+) were 0.45 (±0.1), 0.002 (±0.001) and 0.004 (±0.001) µg mL^−1^, respectively. We also found that reagents typically used to enhance infection were not effective in the CFRK system. This methodology is both efficient and safe; it can be employed by researchers to evaluate neutralizing monoclonal antibodies and contribute to the discovery of new antiviral inhibitors against SARS-CoV-2.

## 1. Introduction

An outbreak of severe acute respiratory syndrome (SARS) from 2002 to 2003 in Guangdong, China, resulted in over 8000 infections and almost 800 deaths. This was caused by a coronavirus (CoV), known as SARS-associated CoV (SARS-CoV) [1]. In December 2019, a novel infectious disease caused by a related coronavirus, SARS-CoV-2, was reported in Wuhan, China. It is widely referred to as coronavirus disease 2019 (COVID-19) [2]. SARS-CoV-2 belongs to the same species as SARS-CoV and other SARS-related bat CoVs [3]. COVID-19 was declared a worldwide pandemic by the World Health Organization (WHO), and has infected over 51 million people and killed more than 1.2 million (December 2020).

SARS-CoV-2 is a β-coronavirus, as characterized by its envelope and non-segmented, positive-sense single-stranded RNA genome [4]. It has a genome size of 29,881 bp, which encodes 9860 amino acids [5]. Two-thirds of the viral RNA encodes two polyproteins and 16 non-structural proteins, whereas the remainder of the viral genome encodes four essential structural proteins: the spike (S) glycoprotein, a small envelope protein, a matrix protein, and a nucleocapsid protein. Several accessory proteins, which interfere with the host innate immune response, are also encoded within the viral genome [4,6,7].

The S protein is 1273 amino acids in length and is coated with polysaccharide molecules, providing camouflage that enables the virus to evade the host immune system during entry [8]. During its biosynthesis, the S protein is cleaved by a furin-like protease in the Golgi apparatus into two subunits, S1 and S2, which form the bulbous head and stalk regions of the S protein. The S proteins of both SARS-CoV-2 and SARS-CoV bind to the angiotensin-converting enzyme 2 (ACE2) receptor, which they use for cell entry [1,9,10]. Following binding of the S protein to the ACE2 receptor, transmembrane protease serine 2 (TMPRSS2), which is located on the host cell membrane, further cleaves the S2 subunit into S2’ and S2’’ subunits, activating the membrane fusion domain [11].

Vero and VeroE6 cells, which express the ACE2 receptor at high levels, have been used to isolate and study SARS-CoV-2. SARS-CoV-2 is categorized as a biosecurity level (BSL) 3 agent, according to WHO guidelines, which impedes the rapid development of new protocols and medicines related to the virus [12,13,14]. To lower the biosecurity risk of viruses, non-replicative pseudo-typed viruses carrying viral S proteins on their surfaces have been developed; these pseudoviruses can infect host cells but cannot replicate, meaning that they can be dealt with using less stringent BSL-2 facilities [15]. Pseudoviruses have recently been developed for SARS-CoV-2 using human immunodeficiency virus (HIV)-based lentiviral particles, murine leukemia virus-based retroviral particles and the vesicular stomatitis virus (VSV) [15]. VSV pseudo-types expressing the SARS-CoV-2 S protein are currently being used to study viral entry [9]; however, the construction of VSV-based pseudo-typed viruses requires a complex procedure. By contrast, pseudo-typed lentiviral viruses can be produced using a comparatively simple method, involving the co-transfection of host cells with multiple plasmids [10]. Human embryonic kidney 293T (HEK293T) cells are currently the main cell line used for the production of pseudo-typed SARS-CoV-2 and to assay its infectivity due to its high transfection efficiency. However, further improvements to this system could be made [3].

Here, we report the development of a simple assay that is 10-fold more sensitive than the commonly-used HEK293T cell assay, using a pseudo-typed lentivirus coated with SARS-CoV-2 S protein (LpVspike(+)). Using our system, we successfully measured the titer of anti-CoV antibodies required for neutralization. Figure 1 shows an illustration outlining our new system.

## 2. Results

### 2.1. Measuring LpVspike(+) Infectivity in Various ACE2-Expressing Cell Types

Although ACE2-expressing HEK293T cells are typically used with the pseudo-typed lentiviral system, the sensitivity of infectivity assays is low. With the goal of identifying a more sensitive cell line, we measured the infectivity of a SARS-CoV-2-S-coated pseudovirus, or LpVspike(+), in the following ACE2-expressing cell lines: Cos7, TZM-bl, Crandell–Rees feline kidney (CRFK), and Vero cells. The various cell types were infected with 100 µL of viral solution in 96-well plates. The luciferase readings, which indicated pseudoviral infection, were > 10-fold higher in ACE2-expressing CRFK (ACE2–CRFK) cells compared to ACE2-expressing HEK293T (ACE2–HEK293T) cells; ACE2–CRFK cells and ACE2–HEK293T cells produced readings of 8 × 10^4^ and 5 × 10^3^ relative luciferase units (RLU), respectively. A reading of approximately 5 RLU was detected in uninfected control cells (Figure 2).

We also examined LpVspike(+) infectivity in VeroE6/TMPRSS2 cells, which have previously been shown to be 10-fold more sensitive to infection than parental VeroE6 cells [16]. LpVspike(+) infected VeroE6/TMPRSS2 cells less efficiently compared to ACE2–CRFK and ACE2–HEK293T cells (Figure 2). A reading of only 10^3^ RLU was detected when VeroE6/TMPRSS2 cells were infected with 100 µL of viral solution. LpVspike(+) did not appear to infect ACE2-expressing Cos7 cells (Figure 2).

The TZM-bl cell line (also called JC.53bl-13) is a HeLa cell derivative engineered to express CD4, CCR5 and CXCR4, as well as Tat-responsive firefly luciferase reporter genes. These features make TZM-bl cells highly susceptible to HIV-1 infection and enables quantification of viral infection and neutralizing antibody titers [17]. Because TZM-bl already has an integrated copy of the luciferase reporter gene, we constructed a second pseudo-typed lentivirus, named LpVspike2(+); this was essentially the same as LpVspike(+), except it did not contain the luciferase gene. ACE2-expressing TZM-bl cells infected with undiluted LpVspike2(+) produced a reading of 2 × 10^4^ RLU, which was comparable to that of the background cell line that had not been virally infected (1 × 10^4^ RLU; Figure 2). Therefore, the TZM-bl cell line was deemed an unsuitable host for our S-covered pseudo-typed lentivirus system.

### 2.2. The Comparison of LpVspike(+) Infectivity in HEK293T and CRFK Cell Lines with/without ACE2 Transfection

The S protein of SARS-CoV-2 uses ACE2 as its point of entry, and hence requires host ACE2 expression to initiate infection. We confirmed the specificity of this process by examining the infectivity of pseudo-typed lentiviruses containing the SARS-CoV-2 S protein (LpVspike(+)) and those that did not (LpVspike[−]) in four cell lines: CRFK, ACE2–CRFK, HEK293T and ACE2–HEK293T. Neither LpVspike(+) nor LpVspike(−) infected the HEK293T cells. LpVspike(+) and a small amount of LpVspike(−) were able to infect ACE2–HEK293T cells, with LpVspike(+)- and LpVspike(−)-infected ACE2–HEK293T cells producing readings of 2000 and 500 RLU with 50 μL of viral solution, respectively (Figure 3a). These results suggest that LpVspike(−) can infect ACE2–HEK293T cells non-specifically, without binding to the ACE2 receptor.

Neither LpVspike(+) nor LpVspike(−) infected CRFK cells. Whereas LpVspike(+) infected ACE2–CRFK cells (10^4^ RLU with 50 µL of viral solution), LpVspike(−) did not (Figure 3b). These results indicate that only LpVspike(+) can infect ACE2-transfected CRFK cells, indicating a S-ACE2 specific interaction.

### 2.3. Effect of Diethylaminoethyl-Dextran on Infectivity

TZM-bl cells are maximally sensitive to HIV infection in diethylaminoethyl-dextran (Deax)-supplemented medium [17]. We therefore examined the effects of Deax on LpVspike(+) and LpVspike(−) infectivity in CRFK and ACE2–CRFK cells. Both LpVspike(+) and LpVspike(−) infected the CRFK cells in the presence of Deax, producing readings of 10^4^–10^5^ RLU per 50 μL of viral solution (Figure 4a); this was in contrast to CRFK cells without Deax, which were only infected by LpVspike(+). The ACE2–CRFK cells were infected by both LpVspike(+) and LpVspike(−) in the presence of Deax, whereas infection only occurred in the ACE2–CRFK line with LpVspike(+) infection (1 × 10^4^ RLU with 50 μL of viral solution) in the absence of Deax. Deax can therefore induce non-specific LpVspike infection in CRFK cells. Notably, only LpVspike(+) efficiently infected ACE2–CRFK cells, LpVspike(−) could not infect CRFK or ACE2–CRFK cells without Deax (Figure 4b).

The addition of Deax increased the infection efficiency of both LpVspike(+) and LpVspike(−) 1000-fold in CRFK cells. While Deax increased the infection efficiency of LpVspike(+) 10-fold in ACE2–CRFK cells, it increased the infection efficiency of LpVspike(−) 1000-fold. These results suggest that Deax can cause non-specific infections in HEK293T, ACE2–HEK293T, CRFK, and ACE2–CRFK cell lines. Deax was therefore ruled out as a supplement for enhancing LpVspike infection efficiency for our system.

### 2.4. Effect of Polybrene on Infectivity

Hexadimethrine bromide, also known as polybrene, is a cationic polymer capable of enhancing in vitro HIV-1 infection by reducing electrostatic repulsion between virions and sialic acid on the cell surface [18]. Polybrene has been widely used in neutralization and infectivity assays examining HIV and pseudo-typed lentiviruses coated with the SARS-CoV-2 S protein [1,15,19,20]. Therefore, we examined the effect of polybrene in our pseudovirus system. Following LpVspike(+) infection, ACE2–CRFK cells produced a reading of approximately 10^4^ RLU with 50 μL of viral solution both with and without polybrene supplementation, suggesting that polybrene did not provide any enhancement effect (Figure 5b). For both CRFK and ACE2–CRFK cells, no LpVspike(−) infection was detected following polybrene addition (Figure 5a,b). Therefore, polybrene did not induce non-specific infections.

### 2.5. Stability of ACE2 Expression in CRFK Cells and LpVspike(+)

We found that CRFK cells were the most sensitive of the cell lines tested, producing a 10-fold higher RLU reading compared to the HEK293T cells. In this experiment, ACE2 was expressed in CRFK cells using a transient system, as opposed to a stable genome-integrated system. We therefore set out to determine the stability of transient ACE2 expression in CRFK cells, with the goal of optimizing transfection conditions. Following seeding and harvesting, ACE2-transfected CRFK cells were incubated for three time periods: 1 h, 1 day, and 2 days. At each time point, the ACE2-transfected CRFK cells were exposed to LpVspike(+) in 96-well plates, then incubated for a further 48 h. The luciferase reporter measurements were similar for each time point, suggesting that the CRFK cells stably express the ACE2 receptor throughout the 48-h period following transfection with the ACE2 vector (Figure 6a).

We then examined the stability of LpVspike(+) following incubation at 37 °C. After 1 h of incubation, luciferase expression had not changed, indicating that LpVspike(+) was stable (Figure 6b).

### 2.6. Inhibition of LpVspike(+) in ACE2-Expressing CRFK Cells with Plasma and Monoclonal Antibodies

We next assessed whether our LpVspike(+)/ACE2–CRFK cell system could be applied to assess antibody neutralization. We first calculated ID50 of two human convalescent sera, SARS-CoV-2 IgG and SARS-CoV-2 IgM, which were 1:350 (±1:20) and 1:1250 (±1:350), respectively (Figure 7). The luciferase readings at ID50 of LpVspike(+) against IgM and IgG were both approximately 1 × 10^4^ RLU (Figure 7a).

Next, we examined the effect of monoclonal antibodies (mAbs) raised against SARS-CoV-2. The IC50 of the IgG, IgM, and IgA anti-SARS-CoV-2 mAbs against LpVspike(+) were 0.45 (±0.1), 0.002 (±0.001), and 0.004 (±0.001) µg mL^−1^, respectively (Figure 8b). Each of the anti-SARS-CoV-2 mAbs was able to neutralize pseudo-typed lentiviruses coated with SARS-CoV-2 S protein during the infection of ACE2-expressing CRFK cells (Figure 8a,b).

## 3. Discussion

In this study, we found that feline CRFK cells were 10-fold more sensitive to infection with S protein-coated HIV-1 pseudoviruses compared to human HEK293T cells. Given that both pseudoviral S-protein expression and CRFK ACE2 expression were required for infection with LpVspike(+), we conclude that our system reflects the natural SARS-CoV-2 infection process. Moreover, this system was tested by measuring the neutralizing activities of convalescent sera and anti-spike mAbs, thus demonstrating its practicality.

In contrast to the high LpVspike(+) infectivity observed in CRFK cells, the infection rates of Cos7 and Vero cells, which are derived from macaque kidneys, were low. The LpVspike(+) pseudovirus was based on HIV-1, meaning that completion of the infection process following entry into the cytoplasm is dependent on whether the host cellular environment facilitates viral uncoating, reverse transcription, movement of the virus into the nucleus, and integration of the viral genome into chromosomes. Macaque cells possess restriction proteins such as APOBEC3 and TRIM5α/TRIM5CypA, which disrupt the HIV-1 genome and core [21], whereas cat cells express a truncated, non-functional version of Trim5α. CRFK cells may lack both functional APOBEC3 and Trim5α. Although human APOBC3 and Trim5α cannot prevent HIV-1 infection, human Trim5α exhibits weak restriction activity [22].

Our attempts to improve infection efficiency by adding Deax and polybrene were not successful. Deax caused non-specific infection in all combinations of cell types and pseudoviruses tested, regardless of the presence of the S protein or its ACE2 receptor. However, Deax specifically enhanced the infection of TZM-bl cells expressing human CD4/CCR5 with a HIV-1 envelope-coated pseudovirus. Although we do not have a definitive explanation for these results, we propose that Deax may disturb the membranes of both the cell and virus. Another unexpected result is that LpVspike(−) infected ACE2–HEK293T cells, but not ACE2–CRFK cells. ACE2 transfection might therefore perturb the membranes of HEK293T cells, causing non-specific infection.

The neutralization assay developed in this work using modified cat CRFK cells is an efficient and safe method that could be a useful research tool, enabling the examination of the binding mechanism between the SARS-CoV-2 S protein and the ACE2 receptor, as well as the development of antiviral drugs. As demonstrated above (Figure 8), this system could be employed to assess SARS-CoV-2 inhibitors and evaluate the neutralizing activity of convalescent antisera and anti-S protein mAbs; anti-S protein antibodies may be useful in treating critically ill COVID-19 patients. Although vaccine development is progressing at an unprecedented rate, this neutralization assay could still be a useful tool for determining the robustness of the B-cell response elicited by newly developed SARS-CoV-2 vaccines.

## 4. Materials and Methods

### 4.1. Cell Culture

HEK293T cells (American Type Culture Collection (ATCC) CRL 3216), CRFK cells (ATCC CCL-94), African green monkey kidney clone of Vero-E6 (VeroE6) cells (ATCC CRL 1586), monkey kidney fibroblast-like cell lines (Cos7; ATCC CRL 1651) and TZM-bl cells (National Institute of Allergy and Infectious Diseases (NIH) ARP #8129) were cultured in Dulbecco’s Modified Eagle Medium (DMEM) (Fujifilm Wako Pure Chemical Corporation, Osaka, Japan), supplemented with 10% (*v*/*v*) heat-inactivated fetal bovine serum, 2 mM sodium pyruvate (MP Biomedicals Inc., Santa Ana, CA, USA) and 4 mM L-glutamine (Fujifilm Wako Pure Chemical Corporation). Cells were harvested and passaged using trypsin/ethylenediaminetetraacetic acid solution (Nacalai Tesque, Kyoto, Japan). All cell lines were maintained at 37 °C under 5% CO_2_ in a humidified environment.

### 4.2. Generation of Pseudo-Typed Lentivirus with SARS-CoV-2 S Particles

Cloning of all DNA plasmids was conducted by transformation into *Escherichia coli* Stbl3 cells, as described previously [23]. Lentiviral-based pseudo-type viruses were constructed as described previously [24]. In brief, pseudo-type lentiviruses coated with the SARS-CoV-2 S protein harboring the vector 2019-nCov_pcDNA3.1(+)-P2A-eGFP spike (1 µg mL^−1^) was prepared by co-transfecting 5 × 10^5^ cells mL^−1^ of HEK293T cells with the HIV-1 NL4-3 ΔEnvΔ Vpr Luciferase Reporter Vector (1 µg mL^−1^) [25].

Additionally, 200 µL of Opti-MEM and 8 µL of X-Treme HP transfection reagent were added. Transfected HEK293T cells were incubated at 37 °C under 5% CO_2_ for 2 days, after which the supernatant containing pseudo-typed lentivirus particles coated with SARS-CoV-2 S protein was harvested and filtered through a 0.45-µm filter (Millipore, Burlington, MA, USA); 400-µL aliquots were stored in 1.5-mL tubes at −80 °C. The 1 µg mL^−1^ HIV-1 NL4-3 ΔEnv Vpr Luciferase Reporter Vector (pNL4-3. Luc. R-E-) (catalogue number 3418) was kindly provided by Dr. Nathaniel Landau (NIH AIDS Reagent Program, Division of AIDS). The 2019-nCov_pcDNA3.1(+)-P2A-eGFP spike vector (catalogue number MC_0101087) was purchased from Nacalai Tesque from Molecular Cloud.

Pseudo-typed lentiviruses with SARS-CoV-2 S protein without the luciferase gene were also constructed to transfect the TZM-bl cell line. Pseudo-type lentivirus particles coated with SARS-CoV-2 S that harbored the 2019-nCov pcDNA3.1 plasmid (1 µg mL^−1^) or 2019-nCoV pCMV plasmid (1 µg mL^−1^) were generated by co-transfecting HEK293T cells with the pSGΔenv vector (1 µg mL^−1^). The pSGΔenv (catalogue number 11051) vector was kindly provided by John C. Kappes and Xiaoyun Wu (NIH AIDS Reagent Program, Division of AIDS).

### 4.3. Generation of Transient ACE2-Expressing Cell Lines

HEK293T, CRFK, Cos 7, and TZM-bl cells were seeded at concentrations of 2.5 × 10^5^ cells mL^−1^ in six-well plates, using DMEM supplemented with 10% fetal bovine serum, 2% L-glutamine and 2% pyruvate. Cell lines were incubated for 1 day, at which point cell confluency reached 60–80%. The pcDNA3.1+/C-(K) DYK-ACE2 (2 µg mL^−1^) plasmid was transfected into the cell lines with 200 µL of Opti-MEM and 8 µL of X-Treme HP transfection reagent. The pcDNA3.1+/C-(K) DYK-ACE2 plasmid (catalogue number MC_0101086) was purchased from Nacalai Tesque (Molecular Cloud). ACE2-expressing cell lines were seeded at a cell concentration of 2.5 × 10^5^ cells mL^−1^ in six-well plates and incubated at 37 °C under 5% CO_2_ for 1 day. HEK293T cells were seeded in poly-L-lysine-coated 96-well plates (Sumitomo Bakelite Celltight Poly-L-lysine-coated flat bottom 96-well plates, catalogue no: ms-0096L). All other cell types were seeded in standard 96-well plates (Falcon tissue culture plate, 96 wells, flat bottom with low evaporation lid).

### 4.4. Neutralization Assays

ACE2-expressing CRFK cells were seeded into 100 µL of medium to a concentration of 2.5 × 10^5^ cell mL^−1^ in 96-well plates 1 day before infection. Previously harvested samples containing pseudo-typed lentiviral particles coated with the SARS-CoV-2 S protein were thawed. To perform the virus titration, pseudo-typed virus samples were serially diluted two-fold a total of nine times and transferred to a 96-well plate. Control wells contained only supplemented medium and no virus sample. Each dilution was made in triplicate. For infection, 100 µL of each LpVspike(+) or LpVspike2(+) dilution was added to the cell-seeded 96-well plates, which were incubated at 37 °C under 5% CO_2_ for 2 days.

Luciferase activity was measured in HEK293T, CRFK, Cos7, Vero, and TZM-bl cells. The details of the neutralization assays performed in this study have been described previously [26,27]. Briefly, to measure luciferase activity, 50 μL of cell lysate solution (Toyo B-Net, Tokyo, Japan) was added to each well, and the plate was agitated for 15 min. An aliquot of 30 μL of lysate was transferred to a Nunc F96 MicroWell white plate (Thermo Fisher Scientific, Waltham, MA, USA), and luminescent substrate (30 μL) was added to each well. Luciferase activity was measured using a TriStar LB 941 reader (Berthold Technologies, Bad Wildbad, Germany) and MikroWin software. ID50 values were calculated as previously described [28].

The neutralization assays were performed using adjusted viral doses that yielded equivalent infectivity levels for each cell line. The neutralization assays were performed in a 96-well format as described previously. ID50 was reported as either the concentration or sample dilution at which the RLU readout was reduced by 50% compared to the RLU readings measured in the virus control wells (cells plus virus without test sample) after subtracting background RLU values from cell control wells (cells only, no virus or test sample) [29].

To test the stability of ACE2-expression in CRFK cells, ACE2-expressing CRFK cells were seeded to a concentration of 2.5 × 10^5^ mL^−1^ in 96-well plates and incubated for three different time periods: 1 h, 1 day and 2 days. Pseudo-typed viruses were then added to a final concentration of 6000 RLU mL^−1^ to the ACE2-expressing CRFK cells and incubated for 48 h at 37 °C under 5% CO_2_.

To evaluate the stability of LpVspike(+) when incubating at 37 °C, diluted pseudo-typed viruses were incubated in triplicate for 10, 30 and 60 min. To test viral infectivity after incubation at 37 °C, virus samples were added to ACE2-expressing CRFK cells in 96-well plates and incubated for 48 h at 37 °C under 5% CO_2_.

SARS-CoV-2 plasma serum of COVID-19 patients with varying IgM and IgG antibody levels was purchased from RayBiotech (Peachtree Corners, GA, USA). The serum plasma was subjected to serial three-fold dilutions, from an initial dilution of 1:60 down to 1:1,180,980. We added 120 μL of supplemented medium to each well in a 96-well plate, with the exception of the outer edge wells, to which 250 μL of phosphate-buffered saline was added. Next, 180 μL of 1/20- or 1/40-diluted plasma was added to the first row of wells and subjected to three-fold serial dilutions. Control wells contained no plasma, only cells and virus. Then, 60 µL of pseudo-typed virus at a concentration of 6000 RLU mL^−1^ was added to wells containing the diluted plasma. The total volume of the diluted plasma and virus was approximately 180 μL. The plasma-virus mixtures were incubated at 37 °C under 5% CO_2_ for 1 h. After incubation, 150 μL of the mixture was added to the seeded ACE2-expressing CRFK cells in 96-well plates. The plates were incubated at 37 °C under 5% CO_2_ for 2 days, and then the luciferase activity of the samples was measured. ID50 was calculated as described previously [28].

Three anti-SARS-CoV-2 neutralizing mAbs were used, including human IgG (catalogue number E-AB-V1021), IgM (catalogue number E-AB-V1026), and IgA (catalogue number E-AB-V1027) isotypes with clone 8A5 Fab that was elicited using recombinant 2019-nCoV S-trimer Protein (His Tag) as the immunogen. The mAbs were diluted 10-fold three times from 1 to 0.001 µg mL^−1^. The experimental method for testing neutralizing mAbs was the same as the method used for assessing the neutralizing ability of plasma. IC50 values were calculated as described previously [28]. N mAbs were purchased from Elabscience (Houston, TX, USA).

## 5. Conclusions

ACE2-expressing CRFK cells were 10 times more sensitive to viral infection compared to the typically used HEK293T cells. We tested our system with a neutralization assay and found that the pseudo-typed lentiviruses coated with SARS-CoV-2 S protein were neutralized by anti-SARS-CoV-2 IgG, IgM, and IgA mAbs and sera. We also evaluated the influence of Deax and polybrene on infectivity. Deax caused non-specific viral infection in the absence of ACE2 receptors and polybrene did not have a significant impact on infectivity or sensitivity. Our neutralization assay using pseudo-typed lentiviruses coated with SARS-CoV-2 S protein and ACE2-expressing CRFK cells is both easier and safer than the methods currently available and can aid researchers in understanding SARS-CoV-2 pathogenesis as well as discovering new drugs and vaccines.

## Figures and Tables

**Figure 1 pathogens-10-00153-f001:**
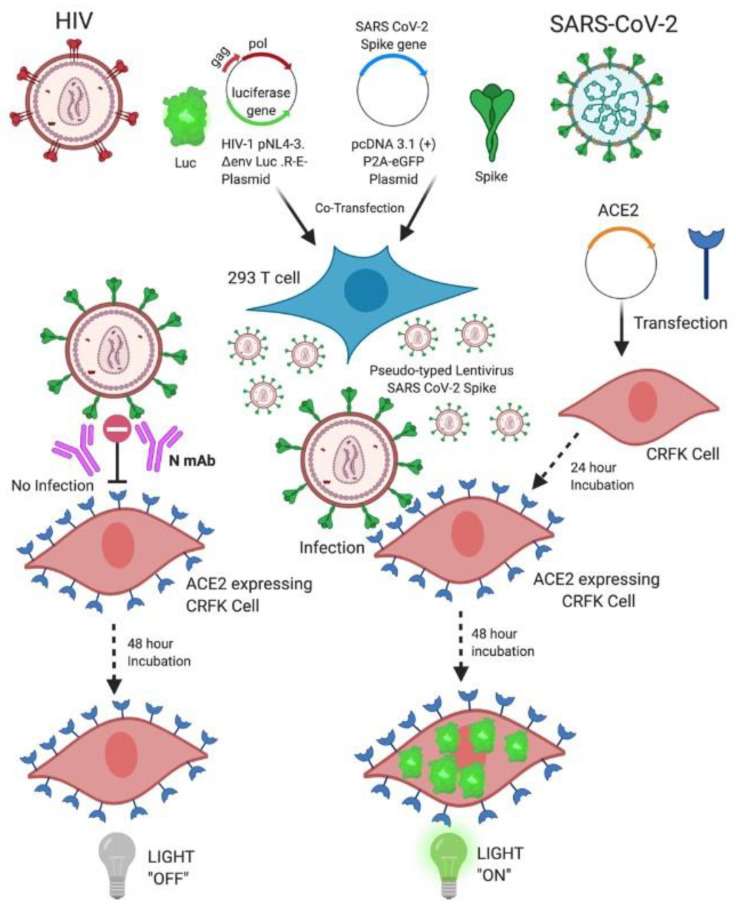
Generation of a pseudo-typed lentivirus coated with severe acute respiratory syndrome coronavirus 2 (SARS-CoV-2) Spike protein using human embryonic kidney 293T (HEK293T) cells, and illustration of the neutralization assay principle using angiotensin-converting enzyme 2 (ACE2)-expressing Crandell–Rees feline kidney (CRFK) cells. The pseudo-typed lentivirus coated with SARS-CoV-2 Spike (LpVspike(+)) and harboring the 2019-nCov_pcDNA3.1(+)-P2A-eGreen Fluorescent Protein (GFP) plasmid was prepared by co-transfecting 5 × 10^5^ cell mL^−1^ of HEK293T cells with the HIV-1 NL4-3 ΔEnv ΔVpr Luciferase Reporter Vector. Transfected HEK293T cells were incubated at 37 °C under 5% CO_2_ for 2 days. The supernatant medium containing LpVspike(+) was harvested and filtered through a 0.45-µm pore size filter. To generate ACE2-expressing cell lines, cells were transfected with pcDNA3.1+/C-(K) DYK-ACE2 and incubated at 37 °C under 5% CO_2_ for 1 day. The transfected cells were then able to transiently express the ACE2 receptor on their membrane surface and could hence be infected by LpVspike(+). Approximately 2.3 × 10^4^ ACE2-expressing cells were infected per well in 96-well plates, and luciferase expression was measured at 48 h post-infection. This figure was created by biorender.com.

**Figure 2 pathogens-10-00153-f002:**
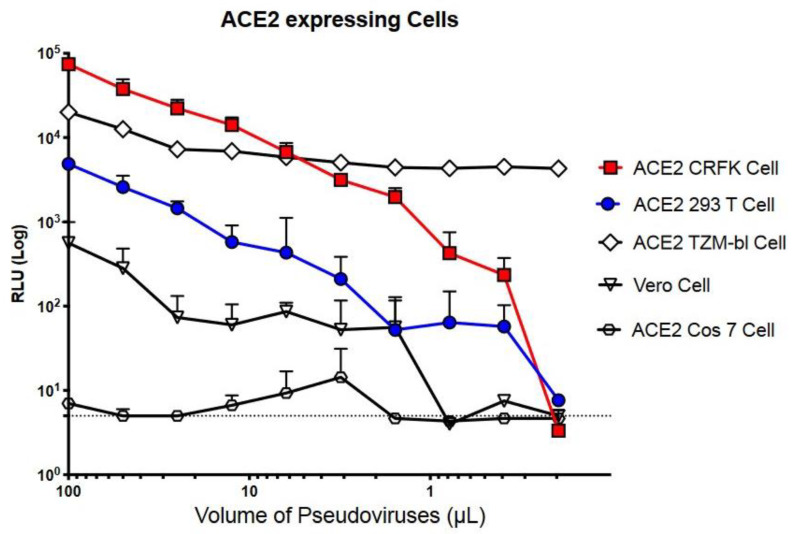
Titers of the LpVspike(+) were determined by measuring relative luciferase units (RLU) at 48 h after infection. Approximately 2.3 × 10^4^ ACE2-expressing cells were seeded per well in 96-well plates. Cells were infected with 100 µL of undiluted LpVspike(+), then two-fold dilutions were made.

**Figure 3 pathogens-10-00153-f003:**
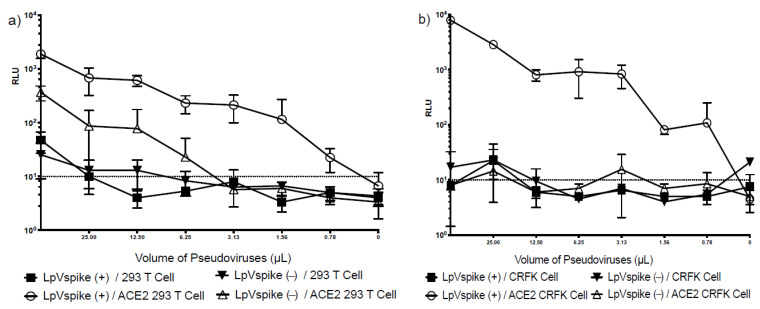
(**a**) Titers of LpVspike(+) and LpVspike(−) as determined by measuring relative luciferase units (RLU) at 48 h after infection in 293 T cell/ACE2 293 T cell. (**b**) Titers of LpVspike(+) and LpVspike(−) as determined by RLU at 48 h after infection in CRFK cell/ACE2-expressing CRFK cell. Cells at concentrations of approximately 2.3 × 10^4^ cell of each cell line (HEK293T, ACE2-expressing HEK293T, CRFK, and ACE2-expressing CRFK) were seeded per well in 96-well plates. The initial volume of LpVspike(+) used for infection was 50 µL of undiluted virus, then three 1:2 dilutions were made.

**Figure 4 pathogens-10-00153-f004:**
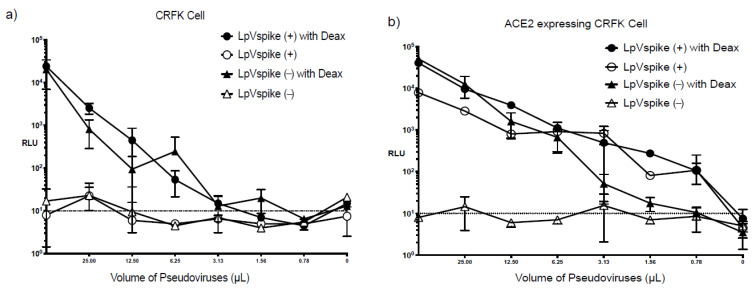
(**a**) RLU measurements were taken at 48 h after infection in CRFK cell. (**b**) RLU measurements were taken at 48 h after infection in ACE2 expressing CRFK cell. Approximately 2.3 × 10^4^ cells of each cell line (CRFK and ACE2-expressing CRFK) were seeded per well in 96-well plates. The initial volume of LpVspike(+) used for infection was 50 µL of undiluted virus, then three 1:2 dilutions were made. Dextran (Deax) was added to the medium to a final concentration of 8 µg mL^−1^.

**Figure 5 pathogens-10-00153-f005:**
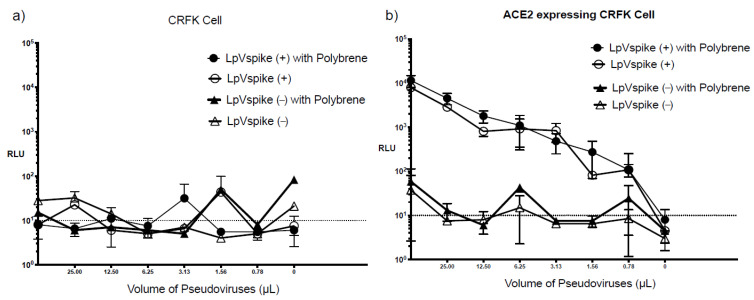
(**a**) RLU measurements were taken at 48 h after infection in CRFK cell. (**b**) RLU measurements were taken at 48 h after infection in ACE2 expressing CRFK cell. Approximately 2.3 × 10^4^ cells of each cell line (CRFK and ACE2-expressing CRFK) were seeded per well in 96-well plates. The initial volume of LpVspike(+) used for infection was 50 µL of undiluted virus, then three 1:2 dilutions were made. Polybrene was added to the medium to a final concentration of 10 µg mL^−1^.

**Figure 6 pathogens-10-00153-f006:**
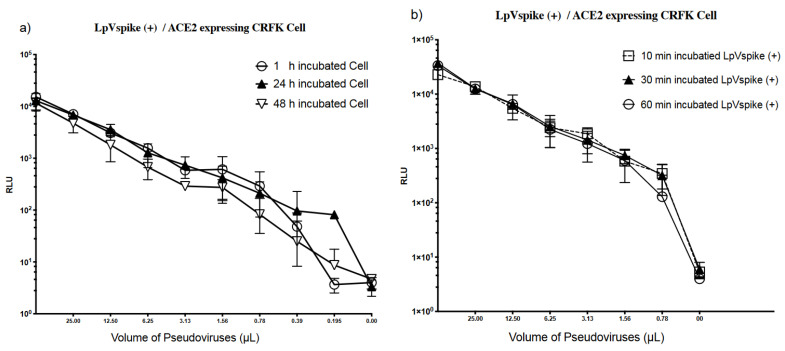
(**a**) Effect of varying the incubation period following ACE2 transfection on LpVspike(+) infectivity. Open circles: infectivity of CRFK cells at 1 h after ACE2 transfection. Black triangles: infectivity of CRFK cells at 24 h after ACE2 transfection. Open inverted triangles: infectivity of CRFK cells at 48 h after ACE2 transfection. The initial volume of LpVspike(+) used for infection was 50 µL of undiluted virus, then three 1:2 dilutions were made. Approximately 2.3 × 10^4^ ACE2-expressing CRFK cells were seeded per well in 96-well plates. Relative luciferase unit (RLU) measurements were taken at 48 h after infection. (**b**) Effect of varying the viral incubation period at 37 °C on LpVspike(+) infectivity. Following infection, cells were incubated with the virus for 10 min (open squares), 30 min (black triangles), or 60 min (open circles). The initial volume of LpVspike(+) used for infection was 50 µL of undiluted virus, then three 1:2 dilutions were made. Approximately 2.3 × 10^4^ ACE2-expressing CRFK cells were seeded per well in 96-well plates. RLU measurements were taken at 48 h after infection.

**Figure 7 pathogens-10-00153-f007:**
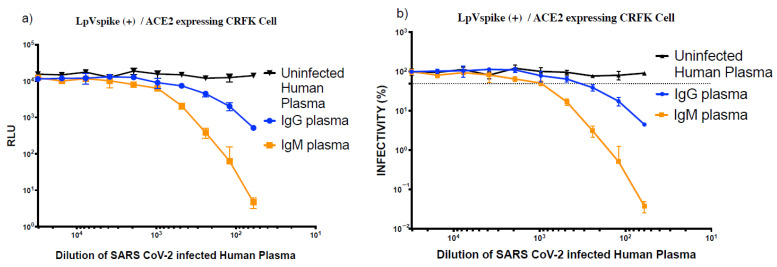
(**a**,**b**) Neutralization of LpVspike(+) with human plasma extracted from patients infected with SARS-CoV-2. After pre-incubating 100 TCID50 of LpVspike(+) with human plasma, the virus-plasma mixture was added to ACE2-expressing CRFK cells and incubated at 37 °C for 48 h, after which luciferase activity was measured. Plasma was subjected to serial two-fold dilutions from an initial dilution of 1:60 down to 1:15,360 (X axis). X axis and Y axis are indicated with log scale. Figure a show row data, figure b is calculated by figure a.

**Figure 8 pathogens-10-00153-f008:**
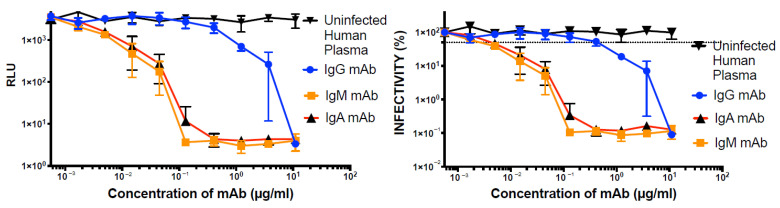
(**a**,**b**) Neutralization of LpVspike(+) with anti-SARS-CoV-2 monoclonal antibodies (mAbs). After pre-incubating 100 TCID50 of LpVspike(+) with each anti-SARS-CoV-2 neutralizing mAb, the mAb-virus mixtures were added to ACE2-expressing CRFK cells and cultured for 48 h, after which luciferase activity was measured. The IgG, IgM, and IgA mAbs were subjected to serial three-fold dilutions, from an initial concentration of 10 µg mL^-1^ down to 0.016 µg mL^−1^. X axis and Y axis are indicated with log scale. Left figure show row RLU data, right figure is calculated by left figure.

## Data Availability

Not applicable.

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
