# Peer review of "A Neutralization Assay Based on Pseudo-Typed Lentivirus with SARS CoV-2 Spike Protein in ACE2-Expressing CRFK Cells"

_pathogens, 2021, doi:10.3390/pathogens10020153_

Round 1
Reviewer 1 Report
The paper wrote by Pisil and collaborators are really well constructed and pleasant to potentials readers. This topic is well explained and methods are really interesting.
I have minor comments needing modifications.
First of all I found the abstract too short and it not explain all methods used and applied by authors and main results as IgG /IgM… sera assays. It need to be improved
In the introduction section, authors highlight the reference 8 as a Spike camouflage but the paper referred as 8 was not well mentioned this aspect. I recommend to replace the reference 8 by this following one “Watanabe Y, Allen JD, Wrapp D, McLellan JS, Crispin M. Site-specific glycan analysis of the SARS-CoV-2 spike. Science. 2020;369:330–3. »
More appropriate to the context and the sentence.
In the result section page 8 authors could add standard deviation in the text (paragraph between figure 7 and figure 8)
In the materials and methods section, but in the whole paper authors have a typo error please modify mg mL.
In the page 8 at the end please modify/rephrase the sentence because its too conclusive “we conclude that our system reflects the natural sars cov2 infection process”
I not really understand why authors could have a negative % of infectivity.? Could authors add information in the figure legend and /or in the materials and methods section.
Author Response
Dear Dr.
Thank you for your kind feedback and recommendations.
I tried to reply to all of them one by one such as following down
Sorry alots, If I did any mistakes when I reply to your questions.
looking forward to hearing from you.
Sincerely
Open Review
(x) I would not like to sign my review report
( ) I would like to sign my review report
English language and style
( ) Extensive editing of English language and style required
(x) Moderate English changes required
( ) English language and style are fine/minor spell check required
( ) I don't feel qualified to judge about the English language and style
Comments and Suggestions for Authors
The paper wrote by Pisil and collaborators are really well constructed and pleasant to potentials readers. This topic is well explained and methods are really interesting.
I have minor comments needing modifications.
- First of all I found the abstract too short and it not explain all methods used and applied by authors and main results as IgG /IgM… sera assays. It need to be improved
- I add main results about IgG, IgM, IgA and sera at the end of the abstract section
- In the introduction section, authors highlight the reference 8 as a Spike camouflage but the paper referred as 8 was not well mentioned this aspect. I recommend to replace the reference 8 by this following one “Watanabe Y, Allen JD, Wrapp D, McLellan JS, Crispin M. Site-specific glycan analysis of the SARS-CoV-2 spike. Science. 2020;369:330–3. »
- I replaced number 8 reference with Watanabe et al, 2020 reference.
More appropriate to the context and the sentence.
- In the result section page 8 authors could add standard deviation in the text (paragraph between figure 7 and figure 8)
- Yes I added standard deviation in the text by
“Severe acute respiratory syndrome coronavirus 2 (SARS-CoV-2) is highly pathogenic zoonotic virus and spread rapidly, with more deaths…………………………………………………………………………………………………………………………………………………………………………………
The ID50 of two human convalescent sera, SARS-CoV-2 IgG and SARS-CoV-2 IgM, which were 1:350(± 1:20) and 1:1250(± 1:350), respectively. The IC50 of the IgG, IgM and IgA anti-SARS-CoV-2 mAbs against LpVspike(+) were 0.45 (±0,1), 0.002 (±0,001) and 0.004 (±0.001) µg mL-1, respectively………………………………”
- In the materials and methods section, but in the whole paper authors have a typo error please modify mg mL.
- I changed mg/mL by µg/mL
- In the page 8 at the end please modify/rephrase the sentence because its too conclusive “we conclude that our system reflects the natural sars cov2 infection process”
- We changed the word“refletcs” by “mimics”
- I not really understand why authors could have a negative % of infectivity.? Could authors add information in the figure legend and /or in the materials and methods section.
- Because this is log figure it is not negative, example 10⁻¹ mean is 0,1 RLU. I explained this situation at the end of legends of figure 7 and 8 by “ X and Y axises are indicated with log scale.
Reviewer 2 Report
The manuscript “A Neutralization Assay Based on Pseudo-typed Lentivirus with SARS CoV-2 Spike in ACE2-expressing CRFK Cells” by Pisil et al. reported a higher-sensitivity neutralization assay using feline CRFK cells, which showed 10-fold higher sensitivity than in standard 293T cells. The manuscript is well-organized, the experiments were done properly, especially the authors provided sufficient information in the material and methods section. I have several comments as listed below.
The neutralization assay itself is not novel, the authors need to avoid using “novel” throughout the manuscript. As the authors also mentioned, this chimera lentiviral/spike system was already developed and used in several studies, therefore, the assay itself is not novel. It will not hurt the importance of the study, but it will be required to remove or substitute “novel” throughout the manuscript. I do agree the finding from this study will advance the field, since the titer of the virus from CRFK cells is 10-fold higher than from 293T cells.
Based on data from the current study, the sensitivity of the chimera virus, LpVspike[+] is 10-fold higher in feline CRFK compared to 293T cells. The chimera viral particles were made from 293T cells, so I was wondering if the viral production would be increased if using CRFK cells, in this case with higher titer of the virus, might observe higher infectivity. If the author has any data or opinion of this, could be added to the results or discussion section.
Minor points/suggestions:
- The title: please add hyphen between “SARS” and “CoV-2” and add “protein” after “SARS CoV-2 spike”, so it will be “SARS-CoV-2 Spike Protein”.
- Results - 2.2. confirmation of S-ACE2 interaction-specific infection- I was expecting to see the biochemistry data, which shows interaction between spike protein and ACE2, although this interaction is already known, however, without showing the biochemical binding data, I would not say “confirmation” in the title, it’s a bit misleading.
- The authors used pNL4-3. Luc. R-E- virus, is there any reason to use Vpr- virus? The standard is to use Env- HIV-1 virus for generating psudo-typed virus, it would help the reader to understand why the authors used both Env and Vpr deleted virus. Please explain in the methods or discussion section. Along the line, in the figure legend 1 and material and methods, please add delta in front of Vpr as well, NL4-3 DEnv D
- Figure 7 and 8, I would assume the right panel figure is from the left panel RLU measurement, however, the author didn’t explain where these numbers are coming from, please add the explanation to the figure legend.
- Figure 7, please explain why at the higher dilution rates, RLU is higher in IgG and IgM plasma than uninfected human plasma.
Author Response
Dear Dr.
Thank you a lot for your kind feedback and recommendations
I tried to reply all of them one by one such as following down
Looking forward to hearing from you
Sincerely
(x) I would not like to sign my review report
( ) I would like to sign my review report
English language and style
( ) Extensive editing of English language and style required
( ) Moderate English changes required
( ) English language and style are fine/minor spell check required
(x) I don't feel qualified to judge about the English language and style
Yes |
Can be improved |
Must be improved |
Not applicable |
|
Does the introduction provide sufficient background and include all relevant references? |
(x) |
( ) |
( ) |
( ) |
Is the research design appropriate? |
(x) |
( ) |
( ) |
( ) |
Are the methods adequately described? |
(x) |
( ) |
( ) |
( ) |
Are the results clearly presented? |
( ) |
(x) |
( ) |
( ) |
Are the conclusions supported by the results? |
( ) |
(x) |
( ) |
( ) |
Comments and Suggestions for Authors
The manuscript “A Neutralization Assay Based on Pseudo-typed Lentivirus with SARS CoV-2 Spike in ACE2-expressing CRFK Cells” by Pisil et al. reported a higher-sensitivity neutralization assay using feline CRFK cells, which showed 10-fold higher sensitivity than in standard 293T cells. The manuscript is well-organized, the experiments were done properly, especially the authors provided sufficient information in the material and methods section. I have several comments as listed below.
- The neutralization assay itself is not novel, the authors need to avoid using “novel” throughout the manuscript. As the authors also mentioned, this chimera lentiviral/spike system was already developed and used in several studies, therefore, the assay itself is not novel. It will not hurt the importance of the study, but it will be required to remove or substitute “novel” throughout the manuscript. I do agree the finding from this study will advance the field, since the titer of the virus from CRFK cells is 10-fold higher than from 293T cells.
- We deleted “novel” from all paper.
- Based on data from the current study, the sensitivity of the chimera virus, LpVspike[+] is 10-fold higher in feline CRFK compared to 293T cells. The chimera viral particles were made from 293T cells, so I was wondering if the viral production would be increased if using CRFK cells, in this case with higher titer of the virus, might observe higher infectivity. If the author has any data or opinion of this, could be added to the results or discussion section.
- Sorry but we don’t have any data or opinion for that
Minor points/suggestions:
- The title: please add hyphen between “SARS” and “CoV-2” and add “protein” after “SARS CoV-2 spike”, so it will be “SARS-CoV-2 Spike Protein”.
- We added “protein” after spike
- Results - 2.2. confirmation of S-ACE2 interaction-specific infection- I was expecting to see the biochemistry data, which shows interaction between spike protein and ACE2, although this interaction is already known, however, without showing the biochemical binding data, I would not say “confirmation” in the title, it’s a bit misleading.
- We change that sentence by 3.2. The comparison of LpVspike(+) infectivity in HEK293T and CRFK cell lines with/without ACE2 transfection
- The authors used pNL4-3. Luc. R-E- virus, is there any reason to use Vpr- virus? The standard is to use Env- HIV-1 virus for generating psudo-typed virus, it would help the reader to understand why the authors used both Env and Vpr deleted virus. Please explain in the methods or discussion section. Along the line, in the figure legend 1 and material and methods, please add delta in front of Vpr as well, NL4-3 DEnv D
- We changed by HIV-1 NL4-3 ΔEnvΔ Vpr Luciferase Reporter Vector
- Actually, this vector also has been using generally on the world.
- Figure 7 and 8, I would assume the right panel figure is from the left panel RLU measurement, however, the author didn’t explain where these numbers are coming from, please add the explanation to the figure legend.
- We add this sentence end of the legends 7 and 8 by “Left figure show row data, right figure is calculated by left figure”
- Figure 7, please explain why at the higher dilution rates, RLU is higher in IgG and IgM plasma than uninfected human plasma.
- We changed this figure with another experimental figure
Submission Date
07 December 2020
Date of this review
17 Jan 2021 17:31:13